# Immunotherapies and Future Combination Strategies for Head and Neck Squamous Cell Carcinoma

**DOI:** 10.3390/ijms20215399

**Published:** 2019-10-30

**Authors:** Valerie Cristina, Ruth Gabriela Herrera-Gómez, Petr Szturz, Vittoria Espeli, Marco Siano

**Affiliations:** 1Oncology Department, Centre Hospitalier Universitaire Vaudois, 1011 Lausanne, Switzerland; valerie.cristina@chuv.ch (V.C.); gabriela.herrera-gomez@chuv.ch (R.G.H.-G.); petr.szturz@chuv.ch (P.S.); 2Oncology Department, Ente Ospedaliero Cantonale, 6500 Bellinzona, Switzerland; vittoria.espeli@eoc.ch; 3Interdisciplinary Cancer Service—SIC, Hôpital Riviera-Chablais, 1847 Rennaz, Switzerland; 4Faculty of Medicine, University and Unive rsity Hospital of Zurich, 8032 Zurich, Switzerland

**Keywords:** head and neck cancer, immunotherapy, combination treatment

## Abstract

Head and neck squamous cell carcinoma (HNSCC) is often diagnosed at an advanced stage and has a dismal prognosis. Nearly 10 years after the approval of cetuximab, anti-PD1/PD-L1 checkpoint inhibitors are the first drugs that have shown any survival benefit for the treatment on platinum-refractory recurrent/metastatic (R/M) HNSCC. Furthermore, checkpoint inhibitors are better tolerated than chemotherapy. The state of the art in the treatment of R/M HNSCC is changing, thanks to improved results for checkpoint inhibitors. Results for these treatments are also awaited in curative settings and for locally advanced HNSCC. Unfortunately, the response rate of immunotherapy is low. Therefore, the identification of predictive biomarkers of response and resistance to anti-PD1/PD-L1 is a key point for better selecting patients that would benefit the most from immunotherapy. Furthermore, the combination of checkpoint inhibitors with various agents is being currently evaluated to improve the response rate, prolong response duration, and even increase the chances for a cure. In this review, we summarize the most important results regarding immune targeting agents for HNSCC, predictive biomarkers for resistance to immune therapies, and future perspectives.

## 1. Introduction

Head and neck squamous cell carcinoma (HNSCC), including cancer of the oral cavity, pharynx, and larynx, are a pervasive cancer with approximately 700,000 new cases and 350,000 cancer-related deaths globally every year [1]. The risk factors most frequently associated with HNSCC are tobacco use, alcohol consumption, and human papillomavirus (HPV) infection. HPV-positive HNSCC typically arises in the oropharynx in younger men who are nonusers of tobacco and alcohol. This subtype of HNSCC is characterized by a distinct biology and a better prognosis than HPV-negative HNSCC [2].

Most HNSCC patients are diagnosed with locally advanced disease comprising Stages III and IV according to World Health Organization (WHO) classification. Despite advances in multimodality treatments involving surgery, radiotherapy, chemotherapy, and/or targeted systemic treatment, prognosis remains poor. More than 50% of patients with a locally advanced disease relapse even though they have been treated with curative intent [3]. Patients with recurrent or metastatic (R/M) HNSCC have dismal prognosis, with a median overall survival (mOS) of less than one year. For patients that are not candidates for chemotherapy or have a progressive disease after platinum-based chemotherapy, prognosis is even worse, with a one-year survival rate of less than 5% [4].

Cetuximab, a monoclonal antibody (mAb) targeting the epidermal growth factor receptor (EGFR) [5] was first approved for second-line treatment based on a phase II trial [6]. Following the publication of the pivotal EXTREME trial [5], cetuximab for first-line treatment in combination with platinum-based chemotherapy was also approved by the Food and Drug Administration (FDA). Since then, and before immunotherapy, no new therapy for R/M HNSCC had shown benefits for the overall survival (OS) of R/M HNSCC patients. After nearly a decade, the first results on immunotherapy in the form of checkpoint inhibitors were able to show survival benefits [7,8,9]. Despite the rather low response rates, these results were welcomed with great enthusiasm, especially since immunotherapy can induce long-lasting responses.

In this review, we summarize the most important results of immune-targeting agents for HNSCC, predictive biomarkers for resistance to immune therapies, and future perspectives.

## 2. Immune System and Cancer

Immune surveillance is crucial for preventing malignant transformation by allowing for the early elimination of cancer cells by the host immune system: the immune system targets unique antigens that are presented on the tumor-cell surface. As a result of this phenomenon, immunocompromised patients show an increased risk of developing cancers as a consequence of immune escape mechanisms.

Both HPV-positive and -negative HNSCCs belong to cancer types with the highest infiltration by immune cells: high median regulatory T-cell (Treg)/CD8+ T-cell ratio and high levels of CD56 natural killer (NK) cell infiltration correlate with better survival [10]. Cancer cells develop mechanisms, evading the immune effects of T cells. Despite this high infiltration by T cells, several suppressive mechanisms have been identified for HNSCC [11,12,13]:—Alterations or deficiencies of tumor human leukocyte antigen (HLA) class 1 expression (for example, induced by EGFR that is expressed in 80% of HNSCC cases) with the overexpression of antigens causing T-cell tolerance.—Increase in immunosuppressive cytokines (interleukine-10, interleukine-6, TGF (tumor-growth-factor)-β).—Aberrant activation of transcription factors signal transducers and activators of transcription 3 (STAT3) that is linked to interleukine-6 signaling.—Downregulation of T cells through the activation of inhibitory T-cell receptors cytotoxic T-lymphocyte-associated antigen 4 (CTLA-4) and programmed cell death protein 1 (PD-1).

Most recently, the tumor microenvironment (TME) has been of increased interest. as it was demonstrated that tumorigenesis is a process that is not only governed by tumor cells. Indeed, the TME includes tumor cells, stromal cells (like carcinoma-associated fibroblasts (CAFs)), tumor-associated macrophage (TAM), endothelial cells, and inflammatory or immune cells, as well as extracellular matrix and soluble factors (like growth factors, cytokines, chemokines, matrix proteins, and proteases) [14]. TME facilitates tumor growth, invasion, migration, and metastasis [15]. TME stromal cells are genetically altered and differ from normal cells. In CAFs isolated from oral cancer, oncogenes and tumor-suppressor genes are expressed at altered levels [16]. CAFs are a major source of tumor-stimulating substances, such as growth factors, chemokines, cytokines, and proteases. It is assumed that reciprocal communication between tumor cells and CAF promotes formation of the TME, favoring tumor growth and invasion [17]. TAMs upregulate the production of factors involved in tumorigenesis. In oral cancer, the high density of TAM is associated with poor prognosis [18,19].

## 3. Checkpoint Inhibitors

### 3.1. ANTI-PD1/PD-L1 Monoclonal Antibody

PD-1 is a surface transmembrane glycoprotein that belongs to the CD28 receptor family and is expressed on activated T cells (CD4 and CD8+ T cells), while programmed death ligand 1 (PD-L1) belongs to the B7 superfamily and is expressed constitutively at low levels ion antigen-presenting cells (APCs), with increased expression on tumor cells [20,21]. The interaction between PD-1 and PD-L1 negatively regulates the physiological antitumor immune response by decreasing cytokine production and inducing T lymphocyte anergy and apoptosis [11]. The physiological aim is to prevent T-cell overstimulation and autoimmunity, but it leads to the inhibition of the antitumor immune response. As a consequence, drugs binding PD-1 or PD-L1 are able to turn off the inhibitory signal driven by this tumor, thus restoring antitumor immunity.

#### 3.1.1. Nivolumab

Nivolumab is a fully human IgG4 anti-PD1 checkpoint inhibitor. This immunotherapeutic agent was approved by the FDA on November 2016, and by the European Medicines Agency (EMA) on April 2017 for the treatment of platinum-refractory R/M HNSCC, on the basis of phase III trial data, showing an overall survival benefit for this patient population [8]. Indeed, in the pivotal CheckMate 141 phase III study, nivolumab (3 mg/kg every two weeks until disease progression, unacceptable toxicity, or withdrawal of consent) demonstrated a survival benefit in patients with R/M HNSCC, with tumor progression/recurrence within 6 months of platinum therapy vs. standard of care (SoC; docetaxel, methotrexate, or cetuximab, according to investigator’s choice): mOS was 7.5 vs. 5.1 months, respectively, *p* = 0.01; regardless of PD-L1 expression (>1% or <1%) and regardless of tumor HPV status [8,22]. However, the median progression-free survival (mPFS) was 2 months for nivolumab and 2.3 months for SoC, and the overall response rate (ORR) was low: 13.3% for nivolumab and 5.8% for standard of care [22].

Treatment beyond progression was allowed for the experimental arm in CheckMate 141. Among 62 patients who received at least one dose of nivolumab after progression, three patients had a >30% reduction in target lesion size [23]. Nevertheless, treatment with nivolumab beyond formal progression should be considered carefully and only performed in the case of clear clinical benefit in order to avoid overtreatment with immunotherapy, potentially leading to missed opportunities for subsequent therapeutic options [24]. In particular, treatment with nivolumab should be stopped in the case of marked performance status declines due to rapid disease progression. Median OS was slightly worse in patients previously treated with cetuximab than in cetuximab-naïve patients (6.9 vs. 8.1 months, respectively) [25]. Nivolumab was well-tolerated; with fewer grade 3–4 adverse events (AEs) than the SoC: 13.1% vs. 35.1%, respectively. The vast majority of grade 3–4 AEs occurred in the first 6 months of initiating treatment of nivolumab, and the most common acute toxicities of any grade comprised fatigue (14%), nausea (9%), and skin rash (8%) [8,22]. The AEs were evaluated according to the Common Terminology Criteria for Adverse Events, version 4.0 [26].

In CheckMate 141, nivolumab even demonstrated a benefit in terms of quality of life (QoL), which was evaluated through three EORTC questionnaires (QoL Questionnaire Core 30 (QLQ-C30), EORTC Head and Neck Cancer-Specific Module (QLQ-H and N35), and three-level European Quality of Life 5-Dimensional questionnaire (EQ-5D)) at baseline, after 2 months and every six weeks thereafter. At baseline, QoL was similar in both arms. While nivolumab stabilized symptoms and functions, patients in the standard arm had clinically relevant deterioration. Therefore, nivolumab delayed the time to deterioration of patient-reported QoL outcomes among patients with platinum-refractory R/M HNSCC that negatively impacted QoL [27]. Moreover, nivolumab is currently being evaluated in a phase II trial as neoadjuvant therapy in patients with previously untreated resectable oral cavity SCC (NCT03021993).

#### 3.1.2. Pembrolizumab

Pembrolizumab, a humanized anti-PD1 mAb, was the first immunotherapeutic agent showing signs of efficacy in HNSCC. In the phase IB KEYNOTE-012 trial, 60 patients with PD-L1 positive (>1%) R/M HNSCC (38% were HPV-positive and 62% were HPV-negative) were treated with pembrolizumab 10 mg/kg intravenously every two weeks [28]. Treatment was well-tolerated, with 17% of patients having grade 3–4 AEs. ORR was 18% (25% in HPV-positive patients and 14% in HPV-negative patients).

In the KEYNOTE-040 phase III trial, patients with R/M HNSCC that progressed within 3–6 months after platinum-containing multimodality therapy were randomized to receive either pembrolizumab monotherapy (200 mg every three weeks) or SoC (docetaxel, methotrexate, or cetuximab, according to the investigator’s choice). Moreover, the study enrolled patients with R/M disease progressing during or after platinum-based first- or second-line therapy. Updated survival results were recently published: pembrolizumab provided a 20% reduction in the risk of death over SoC in patients with R/M HNSCC. Better than expected survival in the standard arm was observed, probably due to subsequent therapies including anti-PD1 mAb (indeed, 13% of patients received subsequent immunotherapy) [9]. On the basis of the KEYNOTE-012 trial data, the FDA approved pembrolizumab as second-line therapy in August 2016, but this approval is currently under revaluation following the results of the KEYNOTE-040 confirmatory trial [29].

Subgroup analyses were performed in KEYNOTE-040. Better survival benefit was observed in the subgroup of patients with a tumor proportion score (TPS) ≥ 50%, which reflects the proportion of tumor cells expressing PD-L1. On that basis, the EMA approved pembrolizumab for treatment of platinum-refractory PD-L1 TPS ≥ 50% R/M HNSCC.

Furthermore, quality of life questionnaires (EORTC QLQ-C30, EORTC QLQ-H and N35 and EQ-5D) were performed. Similarly, to nivolumab in CheckMate 141, pembrolizumab stabilized symptoms, whereas the investigator’s choice led to clinically meaningful deterioration [30].

In the phase III KEYNOTE-048 trial, pembrolizumab was evaluated as first-line treatment, either as monotherapy or in combination with platinum plus 5-fluorouracil chemotherapy, compared with first-line SoC treatment (EXTREME regimen with a combination of platinum and 5-fluorouracil chemotherapy plus cetuximab). The first results of this study were presented at the ESMO congress in October 2018. In the monotherapy arm, pembrolizumab demonstrated a clear survival benefit in patients with PD-L1 combined positive score (CPS) ≥ 20 (mOS 14.9 vs. 10.7 months, HR = 0.61, 95% CI 0.45–0.83, *p* = 0.0007) and, to a lesser extent, in patients with PD-L1 CPS ≥ 1 (mOS 12.3 vs. 10.3 months, HR = 0.78, 95% CI 0.64–0.96, *p* = 0.0086). The CPS reflects the immunehistochemistry (IHC) of the expressed PDL1 target protein in tumor cells, tumor-infiltrating lymphocytes, and macrophages (assessed with the IHC 22C3 pharmDx assay from Agilent), and could have better predictive value than PDL1 expression on tumor cells alone. However, pembrolizumab did not prolong PFS. ORR was lower in the pembrolizumab arm compared to SoC: 23% vs. 36% for CPS ≥ 20, respectively, and 19% and 35% for a CPS ≥ 1. The safety profile was in favor of immunotherapy with a grade 3–5 AE rate of 17% with pembrolizumab vs. 69% with SoC. In these interim analyses, the combination of pembrolizumab plus chemotherapy was found to be noninferior and superior to SoC for OS (mOS 13.0 vs. 10.7 months, HR 0.77, 95% CI 0.63–0.93, *p* = 0.0034), and similarly for the ORR (36% in both arms). The grade 3–5 drug-related AE rate was 71% with pembrolizumab plus chemotherapy and 69% with SoC. Surprisingly, pembrolizumab plus chemotherapy was described as not superior to SoC in patients with a CPS ≥ 20% and ≥1% (numerical data for these comparisons are not available) [31]. A peer-reviewed publication of the final results is pending.

Phase II KEYNOTE-689 is currently ongoing in patients with previously untreated, resectable, locally advanced (LA) HNSCC to evaluate the addition of pembrolizumab as neoadjuvant and postoperative adjuvant therapy (in combination with adjuvant radiotherapy; NCT03765918).

#### 3.1.3. Atezolizumab

Atezolizumab is an anti-PD-L1 mAb. It has been studied in a phase I trial that enrolled 32 patients with R/M HNSCC (including four with nasopharyngeal cancer) [32]. More than half of the patients were heavily pretreated, as they had received at least two prior lines of therapy. Results were in line with those of nivolumab and pembrolizumab with an ORR of 22%, median duration of response of 7.4 months (range of 2.8–45.8 months), mPFS of 2.6 months, and mOS of 6 months. Response in this early trial showed no association with HPV status or PD-L1 expression level.

A phase III study is currently ongoing with atezolizumab as adjuvant therapy after definitive local therapy in patients with high-risk locally advanced squamous cell carcinoma of the head and neck (NCT03452137) [33].

#### 3.1.4. Durvalumab

Durvalumab is a humanized anti-PD-L1 IgG1 mAb. In the single-arm phase II HAWK study, durvalumab has been evaluated as a monotherapy in the treatment of immunotherapy-naïve R/M HNSCC with high PD-L1 protein expression (≥25% of tumor cells). The primary endpoint was the response rate. Durvalumab demonstrated efficacy with an ORR of 16.2% (95% CI, 9.9–24.4%) in all patients, 29.4% (95% CI, 15.1–47.5%) for HPV-positive, and 10.9% (95% CI, 4.5–21.3%) for HPV-negative patients. The mOS was 7.1 months (95% CI, 4.9–9.9) and mPFS was 2.1 months (95% CI, 1.9–3.7). Durvalumab was well-tolerated with 8.0% Grade 3–4 treatment-related AEs, and none led to death [34].

In an attempt to improve the efficacy of monotherapy in patients with low or absent PD-L1 expression, durvalumab was studied in combination with the anti-CTLA-4 mAb tremelimumab. In the phase II CONDOR study, R/M HNSCC patients with low (<25% of tumor cells) or negative PD-L1 expression and whose disease progressed on or after a platinum-containing regimen, were randomized to receive either durvalumab or tremelimumab, or the combination of both agents. Again, the primary endpoint was response rate: ORR was 9.2% (95% CI, 3.46–9.02%) for durvalumab monotherapy, 1.6% (95% CI, 0.04–8.53%) for tremelimumab monotherapy, and 7.8% (95% CI, 3.8–13.8%) for the combination. Treatments were well-tolerated: grade 3–4 treatment-related AEs occurred in 15% of patients overall, and immune-related AEs occurred in 6% of patients in the combination arm. There was no clear benefit of the addition of tremelimumab compared to single-agent durvalumab in patients with low/negative PD-L1 expression [35].

Results of the phase III EAGLE study were recently presented at the annual meeting of the American Society of Clinical Oncology (ASCO) in May 2019. It evaluated durvalumab monotherapy or the combination with tremelimumab vs. SoC in patients with R/M HNSCC, showing disease progression after platinum-based chemotherapy regardless of PD-L1 status. Durvalumab monotherapy and the combination of durvalumab and tremelimumab failed to improve survival when compared with SoC [36]. Like in KEYNOTE-040, the SoC arm outperformed what was usually observed for SoC arms in previous studies. Safety and tolerability were consistent with previous studies.

Looking at rapid changes with the advent of different anti-PD1 and PD-L1 agents for HNSCC, the ongoing phase III KESTREL study will be interesting. KESTREL (NCT02551159) is investigating the combination of durvalumab and tremelimumab vs. standard of care (EXTREME regimen) in previously untreated R/M HNSCC patients.

#### 3.1.5. Avelumab

Avelumab is a fully human anti-PD-L1 IgG1 mAb selectively binds to PD-L1 and competitively blocks its interaction with PD-1, resulting in the activation of T cells. In a phase IA trial including patients with previously treated solid tumors (no HNSCC patients were included), avelumab demonstrated an acceptable toxicity profile, and a dose of 10 mg/kg was chosen for further development [37].

No data have been published yet on avelumab as a monotherapy in R/M HNSCC patients.

In the field of HNSCC, there was a recent media release regarding the CONFRONT phase I/II trial. It evaluated the safety profile of combining avelumab with daily metronomic doses of cyclophosphamide and a single fraction of radiotherapy at 8 Gy in patients with relapsed metastatic HNSCC. The aim of this approach was to reverse the immune evasion of the tumor through a radiotherapy-induced self-vaccination effect [38].

### 3.2. Combination Immunotherapy

Despite all efforts undertaken so far regarding immune therapies for HNSCC, the percentage of patients that respond to inhibitory checkpoint receptor blockade is still unsatisfactory. Furthermore, the understanding regarding which patients finally respond sufficiently and durably is lacking, and more research is needed in the absence of reliable predictive biomarkers. One way of overcoming resistance and immune escape is to tackle more checkpoints in parallel; another is to enhance antigen presentation and, by doing so, ‘priming’ T cells with other different modalities. Therefore, ongoing trials evaluate immunotherapy in combination with various other treatments (cytotoxic agents, other immunotherapeutic agents, radiation therapy) in order to improve the response rate and prolong the response duration, providing a potential chance for curation with the least possible toxicity. Table 1 summarizes immunotherapeutic agent studied in recurrent/metastatic (R/M) head and neck squamous cell carcinoma (HNSCC).

#### 3.2.1. Combination with Other Checkpoint Inhibitors (mAb)

##### CTLA-4 Blockade

CTLA-4 is another inhibitory checkpoint implicated in T-cell downregulation. It is expressed on the surface of activated cytotoxic T lymphocytes, binding to B7 ligands present on the surface of APCs and suppressing activation following T-cell receptor (TCR) binding of the antigens presented by major histocompatibility complex (MHC) proteins. The inhibitory signal of CTLA-4 competes with the stimulatory CD28 receptor for binding to the B7 ligand [13].

The CTLA-4 and PD-1/PD-L1 pathways are considered to be nonredundant. The combination of anti-CTLA4 and anti-PD1/PD-L1 has already demonstrated n a synergistic effect in the blocking of these two checkpoint inhibitors in metastatic melanoma. This synergistic effect results from the fact that PD-1 and CTLA-4 receptors downregulate T cells by distinct mechanisms [40].

Ipilimumab, an IgG1 anti-CTLA-4 mAb, was developed earlier than PD-1/L1-directed therapies. PD-1/PD-L1 inhibitors were developed thereafter, with the intention of being more specific to have fewer side effects than targeting the CTLA-4 pathway, and because PD-L1 expression is specifically found on tumor cells [13].

As explained above, tremelimumab, another anti-CTLA-4 mAb, was evaluated in combination with durvalumab in the phase II CONDOR study and in the phase III EAGLE study [35,36]. In both studies, the addition of tremelimumab failed to improve ORR and outcome.

The combination of ipilimumab and nivolumab is also currently being evaluated in a phase III study as first-line treatment vs. SoC (EXTREME regimen) in R/M HNSCC (CheckMate 651, NCT02741570). Results of this study are due shortly. Furthermore, the combination of nivolumab and ipilimumab is currently being evaluated in the phase I/II IMCISION trial as a neoadjuvant therapy in patients with previously untreated resectable LA HNSCC (NCT03003637).

##### Lymphocyte Activation Gene 3 (LAG-3) Blockade

LAG-3 is expressed on the surface of activated CD4 and CD8+ T cells, as well as several subtypes of NK and dendritic cells [41,42]. It recognizes MHC class II, and preferentially suppresses T cells responsive to stable complexes of peptide and MHC class II.

An anti-LAG-3 mAb, relatlimab (BMS-986016), is evaluated in a phase I/IIA dose escalation and expansion study, alone and in combination with nivolumab in advanced solid tumors, including an HNSCC cohort (CA224-020 study, NCT01968109). Results are currently not available.

##### Other Targets

Other antibody drugs for solid tumors are being developed and could be tested in the treatment of HNSCC in future. For example, antibodies against the T-cell immunoglobulin and mucin domain 3 (TIM-3), T-cell immunoreceptor with Ig and ITIM domains (TIGIT), and killer cell immunoglobulin-like receptor (KIR).

#### 3.2.2. Combination with Other Immune Modulators

Several chemical compound-based immune modulators have been developed and are in the early stages of phase trials as assessed in combination with immunotherapy to improve outcomes.

##### STAT3 Inhibitor

STAT3 is a transcription factor that contributes to the immunosuppressive tumor microenvironment. Preclinical studies have shown that STAT3 inhibition can increase radiosensitivity [43]. The combination of durvalumab and either danvatirsen (AZD9150, a STAT3 inhibitor) or AZD5069 (a CXC chemokine receptor 2 (CXCR2) inhibitor) was evaluated in a phase IB/II trial (SCORES study, NCT02499328). Part A in this study investigated dose escalation in solid tumors with the STAT3 inhibitor (durvalumab alone or durvalumab in combination with danvatirsen) and the CXCR2 inhibitor (durvalumab alone or in combination with AZD5069). Part B included a dose-expansion cohort for HNSCC, and tested the combination of durvalumab with either danvatirsen (STAT3 inhibitor) or AZD5069 (CXCR2 inhibitor) in PD-L1 pretreated/naïve patients, and as monotherapy with primary-endpoint ORR and disease-control rate (DCR). Results from the dose escalation in solid tumor were reported at the 2017 ESMO Annual Meeting and suggested that it enhanced antitumor activity with the combination of danvatirsen and durvalumab [44]. Preliminary results of the dose-expansion cohort with 38 patients with PD-L1 treatment naïve/pretreated R/M HNSCC treated with the combination of durvalumab and danvatirsen were presented at the 2018 ESMO Annual Meeting. The ORR was 26% with four complete responses and six partial responses. Responses were observed regardless of HPV status or PD-L1 expression. Safety and tolerability were confirmed with manageable and reversible AEs, the most important being an increase of thrombocytopenia and liver enzymes [39].

These data suggest antitumor activity from combining an anti-PD-L1 and a STAT3 inhibitor, and warrant further investigation.

##### CXCR2 Inhibitor

CXCR2 is a receptor for cytokines. It is overexpressed in HNSCC and is implicated in disease proliferation via interleukine-8 signaling [11,45]. The CXCR2 inhibitor AZD5069 was evaluated in combination with durvalumab in the early trial SCORES study as mentioned above. Preliminary results of the dose expansion cohort with 20 patients with PD-L1 treatment naïve/pretreated R/M HNSCC treated with the combination of durvalumab and AZD5069 were presented at the 2018 ESMO Annual Meeting. Unfortunately, unlike results of the STAT3 inhibitor, the addition of AZD5069 did not seem to improve outcome as the ORR was 10% and causally related AEs occurred in 76% of patients [39].

##### IDO1 Inhibitor

Indoleamine 2,3-dioxygenase 1 (IDO1) is a catabolizing enzyme that induces immune tolerance by suppressing T cells through tryptophan depletion and kynurenine accumulation in the local tumor microenvironment. It is associated with a poor outcome in laryngeal SCC [46]. Several IDO1 inhibitors are in clinical development.

Epacadostat is a potent and highly selective oral inhibitor of the IDO1 enzyme. Results of the phase I part from phase I/II trial ECHO-202/KEYNOTE-037 combining epacadostat and pembrolizumab in patients with advanced solid tumors have recently been published, but among the 62 enrolled patients, only two had R/M HNSCC [47]. Preliminary results of the HNSCC cohort of this phase I/II trial were presented at the annual 2017 ASCO congress. Eligible patients had PD-1/PD-L1 naïve R/M HNSCC that had received at least one line of platinum-based chemotherapy. Thirty-eight patients were enrolled, two with R/M HNSCC. In patients that had received one or two lines of treatment, ORR was 34% (2 CR and 8 PR) and ORR was 62% (eight stable diseases). In patients with at least three prior lines of treatment, ORR and DCR were 14% (1 PR) and 43% (2 SD), respectively. Response was observed regardless of HPV status. PFS and biomarker analyses have not been presented so far. The combination was well-tolerated, with 11% of patients presenting grade ≥3 treatment-related AEs [48].

Navoximod is a small-molecule inhibitor of IDO1. It was evaluated in combination with atezolizumab in a phase IB study on advanced solid tumors. A total of 158 patients were included in this study, among which were six patients with HNSCC (three in the dose-escalation cohort and three in the dose-expansion cohort). Specific HNSCC results are not available, though this combination was considered safe. The RR was 9% in the dose-escalation cohort, and 11% in the expansion cohort. Hence, although activity was observed, there was no clear evidence of any benefit from the combination of navoximod with atezolizumab in treating unselected patients [49].

#### 3.2.3. Combination with Chemoradiotherapy

Despite multimodality treatments (including surgery followed by radiotherapy or chemoradiotherapy (CRT), or definitive CRT) routinely used for locally advanced HNSCC, the risk of recurrence is high, and prognosis remains dismal. The addition of immunotherapy to radiotherapy or CRT is currently under investigation with the aim of increasing the response rate and reducing the risk of recurrence.

Results of a safety study of pembrolizumab in combination with weekly cisplatin-based CRT have been presented at the 2017 ASCO Annual Meeting. Twenty-seven patients with Stage III–IVB HNSCC received pembrolizumab once prior to CRT, twice during CRT and, finally, five times after CRT. CRT with weekly cisplatin was chosen because it is potentially less myelosuppressive and to avoid the need for dexamethasone that may dampen immune response. All patients completed the full radiation dose (70 Gy) without significant delay, 85% received at least 200 mg/m2 of cisplatin and 78% completed all planned doses of pembrolizumab. Hence, pembrolizumab in combination with weekly cisplatin-based CRT was safe and did not impair CRT [50]. Interim analysis results of this phase IB study were also presented at the 2018 Annual Meeting of the Society for Immunotherapy of Cancer (SITC). In 34 patients with HPV-positive patients, the final complete response rate based on imaging and surgical biopsy was 85%, and one-year PFS was estimated at 97.5%. Again, the addition of pembrolizumab did not seem to impact the safety of standard chemoradiotherapy [51].

These are encouraging results; however, results from randomized trials are needed in order to confirm whether the addition of immunotherapy increases CRT activity. Several phase III studies are ongoing with different checkpoint inhibitors. The KEYNOTE-412 phase III trial with pembrolizumab plus cisplatin-based CRT vs. CRT alone for locally advanced HNSCC is among them (NCT03040999). The phase III JAVELIN Head and Neck 100 trial is currently evaluating the benefit of combining avelumab treatment with the standard definitive chemoradiotherapy with cisplatin, followed by maintenance of avelumab for one year (NCT02952586). In the ongoing phase III REACH trial, the combination of avelumab, cetuximab, and radiation therapy is evaluated in comparison with CRT in fit patients, and with cetuximab and radiation therapy in unfit patients (NCT02999087).

### 3.3. Oncolytic Virotherapy

In preclinical studies, oncolytic viruses have been found to reduce tumor burden and to prime antitumor immunity as well as overcome resistance to checkpoint inhibitors by broadening neo-antigenome-directed T-cell responses [11,52,53].

Talimogene laherparepvec (T-VEC) is a modified attenuated herpes simplex virus type 1 that can induce antitumor response by replicating in tumor cells and producing granulocyte macrophage colony-stimulating factor (GM-CSF). It has already been approved for treatment of metastatic melanoma as a single agent [54]. T-VEC is currently evaluated in the treatment of R/M HNSCC in the ongoing phase IB/III KEYNOTE-137 (NCT02626000). Preliminary results of 36 treated patients with platinum-refractory R/M HNSCC and injectable disease were presented at the ASCO congress in 2018 with an ORR of 16.7% and a disease control rate of 38.9% [55].

Unfortunately, as the ORR was too similar to that of pembrolizumab monotherapy, phase III will not be initiated.

## 4. Immune-Related Adverse Events

Immunotherapy with immune checkpoint inhibitors is better tolerated than chemotherapy. Nevertheless, some patients develop immune-related AEs (irAEs) that can be serious or even fatal. These irAEs are related to drug-induced immune dysregulation.

IrAEs can appear at every organ site, but usually comprise endocrinopathies (such as thyroid insufficiency, acute hypophysitis, hypopituitarism, primary adrenal insufficiency, hypogonadism, hypercalcemia, and diabetes mellitus) [56], gastrointestinal toxicity, liver toxicity and, less frequently, immune-related pneumonitis. In KEYNOTE-048, irAEs were reported separately as “events of interest” affecting about one-quarter of pembrolizumab-treated patients, with 4% classified as grade 3–5. The most common irAEs of any grade were hypothyroidism (15%), pneumonitis (4%), and infusion-related reaction (3%) [8].

As presented at ASCO 2018, in a prospective study including 114 patients with metastatic HNSCC (unselected for PD-L1 status) that received ant-PD1 therapy, the development of irAEs was associated with superior ORR (30.6% vs. 12.3%), PFS, and OS (12.5 vs. 6.8 months, *p* = 0.003) [57].

## 5. Predictive Biomarkers

The special interest in identifying prognostics as well as predictive markers for treatment responses arises because only a proportion of patients benefit from immune checkpoint inhibitor treatment. Different predictive biomarkers are subject to research, but none of these markers have been validated yet. This review focuses on a selection of potential predictive biomarkers that help to identify patients who respond to immunotherapy in HNSCC.

### 5.1. PD-L1 Expression

The expression of PD-L1 is routinely assessed in HNSCC biopsies by IHC. Though PD-L1 overexpression has been associated with higher response rates and better outcomes in patients treated with immune checkpoint inhibitor in different tumor types, cutoff values are not standardized, and the response to treatment is observed in PD-L1 negative cancers [58]. Studies in lung-cancer tissue suggest overexpression as defined as the percentage of viable tumor cells showing partial or complete membrane staining as compared to all viable tumor cells (tumor proportion score or TPS) being greater than 50% as a stronger predictive biomarker [59]; in HNSCC, different cutoff values for PD-L-1 overexpression have been used in clinical trials (Table 2).

Uncertainty arises from the different antibody kits used [60], but also from the examined tissue: besides differences in PD-L1 expression in different types of cancer tissue, dynamic expression varying in intensity depending on location, prior treatments, and inhomogeneous expression inside tumor tissue [61], it has been suggested that PD-L-1 expression should be assessed in the immune and stromal cells of the tumor microenvironment [59], as well as extratumoral sites such as lymph nodes, blood, and bone marrow [62]. These considerations are taken into account in the CPS expressing PD-L1 positivity of cancer microenvironment as a ratio of the number of all PD-L1 positive cells (including lymphocytes, macrophages, and tumor cells) divided by the total number of tumor cells [63].

### 5.2. HPV and Viral Neoantigen

HPV-positive and -negative HNSCC are pathogenetically different entities with different clinical outcomes and prognoses. Persistent infection with HPV type 16 is an important risk factor for the development of oropharyngeal squamous cell carcinoma, with better prognosis in term of OS when treated with standard chemotherapy [2]. These results correspond to IHC findings of HPV-positive HNSCC showing increased infiltration of CD8+ lymphocytes with the presence of less Treg with a potentially less immunosuppressive tumor microenvironment [64]. Even if treatment has yet been not suited to HPV status, it is to be presumed that immune properties also differ between these two patient groups [65]. Whether high PD-L1 expression on IHC for oropharyngeal cancers is due to HPV-related tumors remains unclear [66]. The expression of checkpoints on tumor cells, tumor-infiltrating lymphocytes, or macrophages can have major implications for immune therapies and their efficacy for HNSCC [31].

Clinical trials investigated a treatment response to checkpoint inhibitors with regard to HPV-positive and -negative HNSCC. While the KEYNOTE-012 evaluating pembrolizumab in HNSCC reported an increased ORR in HPV-positive patients (32% vs. 14%) [28], neither the KEYNOTE-040 trial (pembrolizumab) [9] nor the CheckMate 141 trial (nivolumab) [8] could confirm these findings, suggesting that HPV positivity is not a predictive marker for a response to checkpoint inhibitors in HNSCC.

### 5.3. T-Cell-Inflamed Gene Expression Profile

Besides the expression of PD-L-1 in the TME, the composition of infiltrating immune cells (CD8+ T cells, Treg, NK cells, macrophages) seems to have an impact on treatment response to checkpoint inhibitors [67]. The degree of infiltration of CD8+ T cells has been correlated with improved response to anti-PD-1/PD-L1 agents in melanoma [68], results that could be observed for HNSCC in retrospective studies, showing a correlation between treatment response and degree of infiltration with CD8+ T cells, as well as with the ratio of CD8+T cells/Treg [69].

Studies of T-cell activity in cancer tissue have been developed using gene expression profiling (GEP), a technique allowing to identify an “inflammatory T-cell phenotype” [70,71] as well as the “interferon-γ gene expression signature” assay, identifying the expression of genes of T-cell activation [72]. In the KEYNOTE-012 and KEYNOTE-055 trials, GEP was shown to be a predictive marker for checkpoint inhibitor response with benefits in PFS and OS regardless of viral status [72].

### 5.4. Tumor Mutational Burden

Tumor mutational burden (TMB), a technique measuring the total number of mutations in the tumor genome by whole-exome sequencing (WES), has been shown to be a predictive marker of response to checkpoint inhibitors in different tumors [73]. Higher TMB was associated with better outcomes, with special benefit in patients with high TMB when treated with combined anti-CTLA4/anti-PD-1/PD-L1 as compared to monotherapy treatment (ORR 77% vs. 21%) [74].

In HNSCC, TMB was investigated in a retrospective study of 126 patients treated with anti-PD-1/anti-PD-L1 therapy, showing higher TMB in responders compared to nonresponders (median OS 17.7 vs. 7.1 months, *p* < 0.01) [75]; TMB was higher in smokers. Interestingly, patients with HPV-negative cancers who responded to ICI had a greater OS of up to 20 months in the presence of high TMB compared to 6 months in patients with low TMB [69].

## 6. Conclusions

The standard-of-care treatment for HNSCC is currently changing, thanks to the benefits demonstrated by immunotherapy, such as anti-PD-1/PD-L1 checkpoint inhibitors. The latter became a standard treatment for R/M HNSCC by improving OS and QoL and showing a favorable toxicity profile. Responses to immunotherapy can be long-lasting, even in heavily pretreated patients. Nevertheless, the response rate remains low, and combinations with other immunotherapeutic drugs and other treatments are currently under evaluation. Furthermore, the identification of predictive biomarkers in treatment with checkpoint inhibitors is crucial, and predictive biomarkers have been identified in pivotal trials of pembrolizumab and nivolumab in HNSCC with PD-L1 expression in tumor microenvironments, T-cell activity as assessed by GEP and IFN-γ gene expression signature, as well as TMB. A combination of several markers may be necessary to identify patients that may benefit from checkpoint inhibitors. High costs and restricted access to these tests impede their validation and use in clinical practice.

Besides that, TME, and especially CAF and TAM, were identified within a resistance mechanism for immunotherapy and are increasingly regarded as new potential therapeutic targets. Currently, no TME-targeted therapy is used or investigated in particular for HNSCC treatment, but TAM inhibitors are currently being investigated, alone or in combination with standard therapies, in several solid tumors.

## Figures and Tables

**Table 1 ijms-20-05399-t001:** Immunotherapeutic agent studied in recurrent/metastatic (R/M) head and neck squamous cell carcinoma (HNSCC).

Investigational Arm	Target	Phase (Study Name)	Population	ORR (%) (95% CI)	PFS (Months)	mOS (Months) (95% CI)	AEs G3–G5 Rate (%)
						HR for Death (95% CI)	
**First line**							
*Pembrolizumab*	*PD-L1*	*III (KEYNOTE-048)* [31]	*CPS ≥ 20*	*P: 23.3%*	*P: 3.4*	*P: 14.9*	*P: 17% SoC: 69%*
				*SoC: 36.1%*	*SoC: 5.0*	*SoC: 10.7*	
					*HR = 0.99 (0.75–1.29), p = 0.5*	*HR = 0.61 (0.45–0.83), p < 0.01*	
			*CPS ≥ 1*	*P: 19.1%*	*P: 3.2*	*P: 12.3*	
				*SoC: 34.9%*	*SoC: 5.0*	*SoC: 10.3*	
					*HR = 1.16 (0.96–1.39), p = unknown*	*HR = 0.78 (0.64–0.96), p < 0.01*	
*Pembrolizumab + Chemotherapy*	*PD-L1*	*III (KEYNOTE-048)* [31]	*All*	*P + C: 35.6%*	*P + C: 4.9*	*P + C: 13.0*	*P + C: 71% SoC: 69%*
				*SoC: 36.3%*	*SoC: 5.1*	*SoC: 10.7*	
					*HR = 0.92 (0.77–1.10), p = 0.2*	*HR = 0.77 (0.63–0.93), p < 0.01*	
**Second Line and More (Platinum-Resistant)**							
Nivolumab	PD-L1	III (CheckMate 141) [8,22]	All	N: 13.3%	N: 2.0	N: 7.5	N: 13.1% SoC: 35.1%
				SoC: 5.8%	SoC: 2.3	Soc: 5.1	
Pembrolizumab	PD-L1	III (KEYNOTE-040) [9]	All	P: 14.6% (10.4–19.6)	P: 2.1 (2.1–2.3)	P: 8.4 (6.4–9.4)	P: 13% SoC: 36%
				SoC: 10.1% (6.6–14.5)	SoC: 2.3 (2.1–2.8)	SoC: 6.9 (5.9–8.0)	
						HR = 0.80 (0.65–0.98)	
			TPS ≥ 50	ϕ	ϕ	P: 11.6 (8.3–19.5)	ϕ
						SoC: 6.6 (4.8–9.2)	
						HR = 0.53 (0.35–0.81), *p* < 0.01	
Durvalumab	PD-L1	III (EAGLE) [36]	All	D: 17.9%	ϕ	D: 7.6 (6.1–9.8)	D: 10.1% SoC: 24.2%
						SoC: 8.3 (7.3–9.2)	
				SoC: 17.3%		HR = 0.88 (0.72–1.08), *p* = 0.2	
Durvalumab + Tremelimumab	PD-L1 + CTLA-4	III (EAGLE) [36]	All	D + T: 18.2%	ϕ	D + T:6.5 (5.5–8.2)	D + T: 16.3% SoC: 24.2%
				SoC: 17.3%		SoC: 8.3 (7.3–9.2)	
						HR = 1.04 (0.85–1.26), *p* = 0.76	
Durvalumab	PD-L1	II (HAWK) [34]	PD-L1	16.2% (9.9–24.4)	2.1 (1.9–3.7)	7.1 (4.9–9.9)	8%
			≥25%				
Durvalumab	PD-L1	II (CONDOR) [35]	PD-L1	9.2% (3.5–19.0)	1.9 (1.8–2.8)	7.6 (4.9–10.6)	12.3%
			<25%				
Tremelimumab	CTLA-4	II (CONDOR) [35]	PD-L1	1.6% (0.04–8.5)	1.9 (1.8–2.0)	6.0 (4.0–11.3)	16.9%
			<25%				
Durvalumab + Tremelimumab	PD-L1 + CTLA-4	II (CONDOR) [35]	PD-L1	7.8% (3.8–13.8)	2.0 (1.9–2.1)	5.5 (3.9–7.0)	15.8%
			<25%				
Atezolizumab	PD-L1	I [32]	All	22%	2.6	6.0	
*Durvalumab + Danvartisen*	*PD-L1 + STAT3*	*Ib/II (SCORES)* [39]	*All*	*26% (13.4–43.1)*	*ϕ*	*ϕ*	*80%*

Abbreviations: ORR: overall response rate; PFS: progression free survival; mOS: median overall survival; SoC: standard of care; HR: hazard ratio; AEs: adverse events; CPS: numbers of PD-L1 positive cell (tumor cell, lymphocytes, macrophages) divided by all viable tumor cells × 100; TPS: percentage of tumor cell with membranous PD-L1 expression divided by all viable tumor cells; ϕ: no data available. In italics are data presented at the American Society of Clinical Oncology (ASCO) or ESMO Congress that have not yet been peer-reviewed.

**Table 2 ijms-20-05399-t002:** Difference in PD-L1 expression in clinical trials for HNSCC.

Immune Checkpoint Inhibitor	Target	Study	Phase	PD-L1 Expression	PD-L1 Cutoff	ORR (%) Overall	ORR PD-L1 (+) (%)	OS (Months)
Nivolumab	PD-L1	CheckMate 141	III	TCs	>1%	13.3%	17%	7.5
Pembrolizumab	PD-L1	KEYNOTE-012	I	TPS	>1%	18%	17%	NA
				CPS	>1%		22%	
Pembrolizumab	PD-L1	KEYNOTE-040	III	TPS	>50%	14.6%	26.6%	11.6
				CPS	>1		17.3%	8.4
Pembrolizumab	PD-L1	KEYNOTE-0–48	III	CPS	CPS > 1	ϕ	19.1%	14.9 (monotherapy)
					CPS > 20		23.3%	13 (in comb. with chemotherapy)

TC: tumor cell.

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
