# Peer review of "Immunotherapies and Future Combination Strategies for Head and Neck Squamous Cell Carcinoma"

_ijms, 2019, doi:10.3390/ijms20215399_

Round 1

Reviewer 1 Report

This paper reviews recent immunotherapy using monoclonal antibody targeting immune checkpoint inhibitor leading to downregulate T cell activation in the patient with head and neck squamous cell carcinoma (HNSCC). They summarize epidemiology of HNSCC and describe recent immunotherapy alone and also their combinations with immune modulator compounds, etc.

This is timely and important topics. I have some concern in its present form as follows:

Aim of this Journal is based on molecular studies in biology and chemistry, with a strong emphasis on molecular biology and molecular medicine”. We can learn about most updated clinical results for immunotherapies in this review, most of those are based on "not yet peer-reviewed data" presented at the conference such as ASCO. However, molecular mechanism of immure suppressive molecule and its mechanism are not updated in this review. Therefore, this article needs to be consisted with molecular based medicine with recent information from the original articles in scientific Journals. Citation should be up-dated by more recent and important article throughout the text: e.g., Nature Immunology volume 19, pages1415–1426 (2018) for LAG-3. This part is rather out of the text flow. I felt that this part did not need to be separated as an independent paragraph. Abbreviations should be defined in parentheses the first time. Please check throughout of the text: e.g., mAb is defined in line 92 and also line 129. Even if the drugs are not yet to test for HNSCC, information about newly developing antibody drugs are preferably shown for the future therapy, such as antibody for TIM-3 (T-cell immunoglobulin and mucin domain 3), TIGIT (T cell immunoreceptor with Ig and ITIM domains), and KIR (Killer cell Immunoglobulin-like Receptors). Antibody-based drugs should be clearly distinguished with chemical compound based “immune modulators” in the text (there is no explanation about it).

Author Response

We want to thank you for your comments and important remarks and tried to address them all in our new version of the paper. Moreover a summary can be found at the end of this reply:

Point 1:

''Aim of this Journal is based on molecular studies in biology and chemistry, with a strong emphasis on molecular biology and molecular medicine”... ''most of those are based on "not yet peer-reviewed data" presented at the conference such as ASCO.''

We tried to assure with the editors whether our paper fits into the journal and got a positive reply. It is true that some data were unpublished in the moment we submitted the paper but are published in the meantime and was updated accordingly. 

Point 2

''However, molecular mechanism of immure suppressive molecule and its mechanism are not updated in this review.. Citation should be up-dated by more recent and important article throughout the text''

We tried to update as requested with novel citations regarding more specific mechanisms.

Point 3

''I felt that this part did not need to be separated as an independent paragraph. Abbreviations should be defined in parentheses the first time. Please check throughout of the text: e.g., mAb is defined in line 92 and also line 129.. Antibody-based drugs should be clearly distinguished with chemical compound based “immune modulators” in the text (there is no explanation about it).''

We tried to address this remark accordingly. It is clear that at least some phase I/II data should be available and not every new immune-modulating agent will hit the market. We tried to address the most important ones suggested by you in the respective development. 

We hope our paper still finds your approval. The English was revised by the journals editing service.

We attach a short summary of all the changes requested by you and the second reviewer:

CORRECTIONS DONE :

Lines 12, 36, 46, 78, 79, 127, 134, 136, 148, 163, 167, 215, 272, 405, 422, 444 : Abbreviation correction, as requested by 1st reviewer

Line 19 : Suppression of « (cytotoxic agents, others immunotherapeutic agents, radiation therapy…)” as resquested by 2ndreviewer

Line 21-22 : Concluding statement that summarizes the aim and content of the review, as requested by 2nd reviewer

Lines 27-31 : Few lines on HPV and how it is associated with some forms of HNSCCs, as requested by 2nd reviewer

Lines 37-40 : statement revised, as requested by 2nd reviewer

Lines 40-44 : statement revised, as requested by 2nd reviewer

Lines 59-71 : Sentence modified for ease of reading, as requested by 2nd reviewer

Lines 68 and 107-115 : Sentence has been moved as introduction in the section 3.1, as requested by 2nd reviewer

Lines 68 and 289-293 : Sentence has been moved as introduction in the section 3.2.1.1, as requested by 2nd reviewer

Line 98-104 : 2nd reviewer asked if this is for HPV+ oropharyngeal cancer. The answer is no.

Lines 93-97 : Sentence modified for ease of reading, as requested by 2nd reviewer

Lines 118-124 : Table with summary of data on immunotherpeutic agents studied in R/M HNSCC, as requested bz 2ndreiewer

Line 146 : sentence removed as requested by 2nd reviewer

Line 160-161 : updated data

Lines 175-187 : updated data and updated references

Lines 211-213 : updated data

Lines 225-233 : updated data and updated references

Line 245 : updated reference

Lines 246-256 : updated data and updated references

Lines 268 and 430-433: Sentence on JAVELIN study has been  moved in the section 3.2.4 (combination with chemoradiation)

Line 303 : Reminder of what tremelimumab is, as requested by 2nd reviewer

Lines 303-313 : updated data and updated references

Lines 317-319 : updated data and updated reference, as requested by 1st reviewer

Line 320 : updated data

Lines 323-327 : additional informations on newly developing antibody drugs, as requested by 1st reviewer

Line 287 and 329-330 : additional informations to better distinguish the section on antibody-based drugs and the chemical compound based « immune modulators » , as requested by 1st reviewer

Line 342-356 and 363-368 : updated data and reference

Lines 286-287 and 316-318 : additional informations on these target and biological role, as requested by 2nd reviewer

Line 371-372 : addional information on mechanism of action, as requested by 1st reviewer

Line 386-392 : updated data and reference

Line 394 and 441-453 : subsection on oncologic virotherapy has been moved in a new subsection (3.3) because it was not combined with anti-PD1/PD-L1, as requested by 2nd reviewer

Lines 68 and 500-510 : Sentence has been moved, as requested by 2nd reviewer

Lines 546 and 558-564 : Sentence has been moved, as requested by 2nd reviewer

QUESTIONS FROM THE 2nd REVIEWER :

- « Did all these studies report the response rates between HPV- and + HNSCC patients ? »

No, it was not reported in all this studies. When RR according to HPV status was reported, we mentioned the results in our review.

Kind regards

Reviewer 2 Report

This review summarizes current developments in the use of immunotherapeutic drugs for HNSCCs. Authors provided detailed drug trial and technical accounts for each drug. This review could serve as a valuable resource for clinicians and fellow scientists interested in this work. Given the mountain of information presented here, the current review is frustratingly difficult to follow. The current form reads almost like a technical document laced with an overwhelming amount of data. Furthermore, this document requires major English language proofreading to make it comprehensible to fellow readers. This current review lacks critical analysis of current findings even though their concluding statement somewhat addresses this. Authors are asked to address comments below, which should hopefully transform this review to one that could be read with ease.

Sporadic grammatical/spelling errors found, and several sentences have poor comprehension. Some examples are listed below throughout. Inappropriate usage of comas found throughout the manuscript. English proofreading highly recommended Authors are advised to remove this statement from their abstract “… (cytotoxic agents, others immunotherapeutic agents, radiation therapy...)…”. The abstract should have a concluding statement that summarizes the aim and content of this review Line 28 – Change “although” to “though” Line 30 – 31 – Revise statement. It does not make sense Line 33 – 35 – Revise statement. It does not make sense Line 38 – Do you mean “in the form of”? Authors should dedicate a few lines on HPV and how it is associated with some forms of HNSCCs, especially those of the oropharynx. Authors need to introduce what HPV is etc. Line 50-58 – Sentence is too long and verbose. Please refine for ease of reading Line 61-63 – This sentence should be moved after introducing PD-1 in its respective subsection Line 65-78 – This information would suit better if it is introduced in its respective subsection i.e. 3.1 and 3.2.1.1. Given the central PD-1 theme in this section, subsection 3.1 would benefit with a short background on PD-1 Line 71-72 – Do you mean “As a consequence”? Line 80-83 - Sentence is too long and verbose. Please refine for ease of reading Lines 85 and 90 – Is this for HPV+ oropharyngeal cancer? Please be precise Line 118-119 – Remove this statement. It is unnecessary Line 166 – Remove this sentence A table summarizing all the therapeutic agents described in this review is highly recommended to include information such as treatment success rate, treatment duration etc. The current way this review reads makes it hard for fellow readers to grasp key information Did all these studies report the response rates between HPV – and + HNSCC patients? If so, authors are encouraged to mention this. Authors have only described this for Pembrolizumab. This is important as HPV+ HNSCC patients have better response rates that HPV- patients with current treatments Section 3.2.1.1, Line 242 – Authors need to remind readers what tremelimumab is. This term was mentioned much earlier in the review Lines 249, 256, 274– Authors need to provide a brief background for these targets and its biological role Line 305 – Did the KEYNOTE-137 trial involve combination of T-VEC with anti-PD1/PD1L agents? This section is looking more like a monotherapy than a combination therapy Line 408-413 – This paragraph is better suited in the Conclusion section

Author Response

Wa want to thank the second reviewer for his important remarks;

Point 1

''.. this document requires major English language proofreading to make it comprehensible to fellow readers. This current review lacks critical analysis of current findings even though their concluding statement somewhat addresses this. Authors are asked to address comments below, which should hopefully transform this review to one that could be read with ease.''

An English professional editing and review was done by the service recommended by the journal. We tried to rewrite the text accordingly, to make the paper more understandable and easier to read.

We addressed the corrections and requests according to your suggestions, hoping to find your approval for publication.

We attach a short summary of all the changes requested by you and the first reviewer:

CORRECTIONS DONE :

Lines 12, 36, 46, 78, 79, 127, 134, 136, 148, 163, 167, 215, 272, 405, 422, 444 : Abbreviation correction, as requested by 1st reviewer

Line 19 : Suppression of « (cytotoxic agents, others immunotherapeutic agents, radiation therapy…)” as resquested by 2ndreviewer

Line 21-22 : Concluding statement that summarizes the aim and content of the review, as requested by 2nd reviewer

Lines 27-31 : Few lines on HPV and how it is associated with some forms of HNSCCs, as requested by 2nd reviewer

Lines 37-40 : statement revised, as requested by 2nd reviewer

Lines 40-44 : statement revised, as requested by 2nd reviewer

Lines 59-71 : Sentence modified for ease of reading, as requested by 2nd reviewer

Lines 68 and 107-115 : Sentence has been moved as introduction in the section 3.1, as requested by 2nd reviewer

Lines 68 and 289-293 : Sentence has been moved as introduction in the section 3.2.1.1, as requested by 2nd reviewer

Line 98-104 : 2nd reviewer asked if this is for HPV+ oropharyngeal cancer. The answer is no.

Lines 93-97 : Sentence modified for ease of reading, as requested by 2nd reviewer

Lines 118-124 : Table with summary of data on immunotherpeutic agents studied in R/M HNSCC, as requested bz 2ndreiewer

Line 146 : sentence removed as requested by 2nd reviewer

Line 160-161 : updated data

Lines 175-187 : updated data and updated references

Lines 211-213 : updated data

Lines 225-233 : updated data and updated references

Line 245 : updated reference

Lines 246-256 : updated data and updated references

Lines 268 and 430-433: Sentence on JAVELIN study has been  moved in the section 3.2.4 (combination with chemoradiation)

Line 303 : Reminder of what tremelimumab is, as requested by 2nd reviewer

Lines 303-313 : updated data and updated references

Lines 317-319 : updated data and updated reference, as requested by 1st reviewer

Line 320 : updated data

Lines 323-327 : additional informations on newly developing antibody drugs, as requested by 1st reviewer

Line 287 and 329-330 : additional informations to better distinguish the section on antibody-based drugs and the chemical compound based « immune modulators » , as requested by 1st reviewer

Line 342-356 and 363-368 : updated data and reference

Lines 286-287 and 316-318 : additional informations on these target and biological role, as requested by 2nd reviewer

Line 371-372 : addional information on mechanism of action, as requested by 1st reviewer

Line 386-392 : updated data and reference

Line 394 and 441-453 : subsection on oncologic virotherapy has been moved in a new subsection (3.3) because it was not combined with anti-PD1/PD-L1, as requested by 2nd reviewer

Lines 68 and 500-510 : Sentence has been moved, as requested by 2nd reviewer

Lines 546 and 558-564 : Sentence has been moved, as requested by 2nd reviewer

QUESTIONS FROM THE 2nd REVIEWER :

- « Did all these studies report the response rates between HPV- and + HNSCC patients ? »

No, it was not reported in all this studies. When RR according to HPV status was reported, we mentioned the results in our review.

Kind regards

Round 2

Reviewer 1 Report

The author have addressed all my concern. However, following are needed to be revised (mainly about format or style).

Line 64: Use Greek letter for beta.

Line 177: Use en dash “–” not hyphen “-” between 0.63 and 0.93

Line 209: Change “3.46-19.02%” to “3.46%–19.02%”.

Table 1: Change “0.96-1.39” to “0.96–1.39”.

Line 209: Change “3.46-19.02%” to “3.46%–19.02%”.

There are mixed “versus” (Line 108, 175, 365, 461) and “vs.” (Line 109, 121, 123, 167, 168, 173, 177, 218, 225, 268, 399, 440, 464), likewise, PD-L1 and PDL-1.

References: Please change all the list to style of IJMS according to “Instructions for Authors”. Use en dash “–” not hyphen “-” between page numbers.

Table 1: Please add citation number to each study.

Author Response

Thank you again for the second review and the attentive remarks we were able to adopt all:

Line 64: Use Greek letter for beta. ; done

Line 177: Use en dash “–” not hyphen “-” between 0.63 and 0.93; done

Line 209: Change “3.46-19.02%” to “3.46%–19.02%”. ; done

Table 1: Change “0.96-1.39” to “0.96–1.39”. ; done

Line 209: Change “3.46-19.02%” to “3.46%–19.02%”. done 

There are mixed “versus” (Line 108, 175, 365, 461) and “vs.” (Line 109, 121, 123, 167, 168, 173, 177, 218, 225, 268, 399, 440, 464), likewise, PD-L1 and PDL-1.; both adapted and harmonized

References: Please change all the list to style of IJMS according to “Instructions for Authors”. Use en dash “–” not hyphen “-” between page numbers. ; done

Table 1: Please add citation number to each study. ; done

Kind regards